# S-Bend and Y Waveguide Architectures in Germanate Glasses Irradiated by Femtosecond Laser

**DOI:** 10.3390/mi16020171

**Published:** 2025-01-31

**Authors:** Thiago Vecchi Fernandes, Camila Dias da Silva Bordon, Niklaus Ursus Wetter, Wagner de Rossi, Luciana Reyes Pires Kassab

**Affiliations:** 1Departamento de Engenharia de Sistemas Eletrônicos, Escola Politécnica da Universidade de São Paulo, Av. Prof. Luciano Gualberto, 158, Travessa 3, São Paulo 05508-010, SP, Brazil; camiladsb@usp.br; 2Instituto de Pesquisas Energéticas e Nucleares, IPEN-CNEN, 2242, Av. Prof. Lineu Prestes, São Paulo 05508-000, SP, Brazil; nuwetter@ipen.br (N.U.W.); wderossi@ipen.br (W.d.R.); 3Faculdade de Tecnologia de São Paulo, CEETEPS, Praça Cel. Fernando Prestes, 30, São Paulo 01124-060, SP, Brazil; kassablm@fatecsp.br

**Keywords:** waveguides, GeO_2_–PbO glasses, S-bend, Y-shaped waveguides

## Abstract

This study is focused on the fabrication and characterization of various dual waveguides through femtosecond (fs) laser irradiation of GeO_2_-based glass samples. The objective of the present work is to develop diverse waveguide configurations, namely straight, S-bend and Y-shaped waveguides within GeO_2_–PbO glasses embedded with silver nanoparticles, utilizing a double-guide platform, for photonic applications such as resonant rings and beam splitters. Enhanced guidance was observed with a larger radius of curvature (80 mm) among the two distinct S-bend waveguides produced. The maximum relative propagation loss was recorded for the S-bend waveguide with a 40 mm radius, while the minimum loss was noted for the Y-shaped waveguide. In the latter configuration, with an opening angle of 5° and a separation of 300 µm between the two arms, an output power ratio of 50.5/49.5 between the left and right arms indicated promising potential for beam splitter applications. During the study, the quality factor (M^2^) of the proposed architectures was measured and the 80 mm S-bend configuration presented the best symmetry between the x and y axes; in the case of the Y configuration the similarity between the M^2^ values in both axes, for the first and second arms, indicates comparable light guidance.

## 1. Introduction

In recent years, the telecommunications industry has undergone significant expansion due to the rising demand for internet connectivity. The increase in data processing speed and the proliferation of wireless internet devices have required bandwidth growth, leading telecommunications companies to explore more efficient data transmission technologies. In this context, the field of optical telecommunications has gained prominence, with waveguides meeting the essential criteria for advanced applications.

The incorporation of monolithic waveguide-based laser sources into compact, small-cavity photonic devices has opened new prospects for integrated optics [1]. This includes developments such as Q-switched waveguide lasers [2], compact solid-state waveguide lasers [3] and dual-doped waveguides employing femtosecond (fs) laser writing [4], among others. Waveguides, similar to other fundamental photonic components, confine light within confined volumes, thus increasing the intensity within the waveguide cavity relative to the surrounding material [1,5].

Heavy metal oxide glasses have emerged as viable alternatives in this field, offering mechanical and optical properties that are comparable to, or even surpass, those of traditional silica optical fibers. Notably, germanate glasses are particularly distinguished in this regard. GeO_2_-PbO glasses, for instance, possess several advantageous attributes: a low phonon energy (800~975 cm^−1^), a broad transmission window (400~4500 nm), a high refractive index (2.0) due to the high atomic mass of the constituent elements and their high polarizability, as well as favorable chemical, thermal and mechanical stabilities. Additionally, these glasses have a lower melting point compared to silicate glasses, a lower glass transition temperature and good solubility for rare earth ions (TRs) [6,7].

Doped GeO_2_-PbO glasses with TRs have shown improved optical properties, owing to the plasmonic effects of metallic nanoparticles (NPs) [8]. Furthermore, these glasses with metallic NPs exhibit significant potential for photonic applications due to their ultrafast response times and high third-order nonlinearities [9]. Initial results from femtosecond laser-written dual-line waveguides directly inscribed in GeO_2_ and TeO_2_-based glasses were reported in [10]. More recently, we demonstrated a novel configuration of dual-line waveguides produced in Nd^3+^-doped GeO_2_-PbO glass without and with metallic nanoparticles, aimed at optical amplifiers operating at 1064 nm [11,12,13]. It is important to highlight that enhanced optical gain at 1064 nm was demonstrated due to the plasmonic effects of Ag and Au nanoparticles [12,13].

Building on these promising results, which indicated enhanced beam quality factors and optical gain, we present, for the first time, the fabrication and characterization of dual-line waveguides with various configurations including straight, S-bend [1,14] and Y-shaped [15,16]—using GeO_2_-PbO glasses with silver (Ag) NPs. These waveguides were inscribed directly into the glasses via femtosecond (fs) laser, targeting applications in photonic devices such as resonant rings [17], high-gain amplifiers [1] and beam splitters [18], among others.

We utilized a Ti:Sapphire (fs) laser operating at 800 nm, which delivered 30 fs pulses at a 10 kHz repetition rate and pulse energy of 30 µJ. We present results of beam quality factor M^2^ (at 632 and 1064 nm), propagation loss, polarization and output power ratio between the left and right arms of Y-shaped waveguides.

This study addresses a gap in the literature, as there are limited reports on waveguides inscribed in glasses with Ag nanoparticles using fs lasers [15]. With the growing demand for S-bend waveguides, research on this topic has become increasingly prevalent. As a result, studies on the fabrication and characterization of straight and S-bend channel waveguides in crystalline hosts [1] via direct fs laser writing have been identified in the literature. Recently, results were reported on optical waveguides with curved and straight configurations in a crystalline silicon wafer and in monocrystalline silicon using a nanosecond laser irradiation [19,20]. Moreover, results in glasses with an S-bend have been reported in silica glass [14]. Glass substrates such as phosphates [15] and fused silica [16] have been documented regarding Y-shaped architectures. The relatively sparse research on germanate glasses has thus inspired this investigation. The examination of this material’s applications and the clarification of its optical properties are crucial for the development of new photonic devices. Furthermore, the advantages of fs laser techniques for manufacturing waveguides are noteworthy, as opposed to conventional methods involving thin films and silicon technology in clean room environments that present a higher level of complexity due to more steps of the process.

## 2. Materials and Methods

### 2.1. Preparation of the Glasses

The glass manufacturing process employed the base composition of 40 wt% GeO_2_ and 60 wt% PbO (designated as GP), with the incorporation of 2.0 wt% AgNO_3_. The samples were synthesized using the melt-quenching method. This involved the fusion of the constituent materials in a high-purity alumina crucible (99.999%) at a temperature of 1200 °C, with a glass rod utilized for thorough homogenization. Following this, the molten glass was quenched into preheated brass molds and subjected to annealing at 420 °C for 1 h. This annealing step is essential for alleviating internal stresses and enhancing the durability of the glass samples. Post-annealing, the samples were gradually cooled to ambient temperature within the oven. Subsequently, a final polishing procedure was carried out to achieve the desired surface finish [12].

### 2.2. Waveguide Manufacture

Double-walled waveguides were generated, consisting of a pair of parallel walls, each one created through the superposition of four tracks generated by overlapping laser pulses. Due to spherical aberration and nonlinear self-focusing effects, these tracks have thin, high walls, of approximate dimensions of 180 μm in height and 3 μm in width when observed through the microscope, and that show a decreased refractive index (refractive index change of 10^−3^) [21], allowing guidance only in the region between of these walls.

These tracks were created using a fs laser system operating at 800 nm (Ti:sapphire, Femtopower Double 10 kHz/Femtolasers Produktions GmbH, Vienna, Austria) equipped with a focusing lens of 20 mm focal length and numerical aperture (N.A) of 0.23, with the focal point positioned 0.75 mm beneath the sample surface. The laser beam was oriented perpendicularly to the polished surface of the sample, with linear polarization at an angle of 45° relative to the direction of movement, operating at a speed of 0.5 mm/s and a pulse energy of 30 μJ. Previous measurements indicated that polarization at 45° is more favorable and also easier from a practical point of view of our experimental setup.

The study involved the creation of different types of waveguides: Straight, S-bend and Y waveguides. The straight waveguide presents the configuration comprising two parallel lines, separated by a distance of 10 µm. The S-bend waveguides produced are separated by a distance of 10 µm, already used in previous reports [11,12,13] and bending radii of 40 and 80 mm. This choice was based on the results already reported [21]. For example, S-bend waveguides of 5 and 10 mm did not demonstrate light guidance and 20 mm showed high propagation loss and M^2^ value. The Y-shaped waveguides featured an opening angle of 5° and a 300 µm separation between its arms while maintaining a distance of 10 µm between the guide walls. Previous experiments with Y-shaped waveguides [21] showed that opening angle of 5° and a 620 µm separation between its arms led to a worse performance. Figure 1 illustrates the sequence from left to right of the waveguides produced: one straight, two S-bend and the Y-shaped.

The focused femtosecond (fs) laser causes localized microscopic modifications in materials that can be catastrophic, leading to the breaking of chemical bonds and even local melting and evaporation of the material. Less severe changes can alter the local crystalline arrangement due to the high intensity of the laser pulses and thus modify the refractive index in the affected region. Careful selection of process parameters, such as intensity and number of overlapping pulses, can lead to an irreversible and permanent modification of this refractive index change [5,22,23,24].

Figure 2 demonstrates the configuration used to write the waveguide, as well as the damage at the ends of the sample. It is worth noting that the distance between two pairs of waveguides is 400 µm. After writing, the entry and exit faces normally show superficial damage, which requires new polishing.

Figure 2 also shows that each waveguide is formed by the micromachined regions, thus forming two parallel walls with visible dimensions of 178 µm (height) × 3 µm (width) and 10 µm separation in between. Due to their dimensions and the minimal contrast with the surrounding medium, these regions are challenging to observe and can only be visualized under polarized light microscopy and specific conditions.

### 2.3. Characterization

Using the setup depicted in Figure 3a, the propagation losses for all waveguide configurations were assessed at a wavelength of 632 nm, with the straight waveguide having a 10 µm separation serving as the reference [25]. The output power distribution was measured for the two S-bend waveguides (with 40 and 80 mm radius) and the two arms of the Y-shaped waveguide. Equation (1) was utilized to determine the relative propagation losses [25].(1)$$\eta =-10\log_{10}\left(\frac{\sum_{i}P_{i}}{P_{0}}\right)$$

In the equation above, *η* denotes the value of the relative losses in decibels (dB), $P_{i}$ represents the sum of the output powers of the waveguide being compared, and $P_{0}$ signifies the output power of the 10 µm waveguide used as a reference. The standard method for measuring propagation losses is the cutback method [11,12,13,26]. However, when dealing with S-bend and Y-shaped architectures, cutting the sample is not feasible. In such cases, to circumvent the need for sample cutting, the relative losses procedure is employed.

The M^2^ was measured for all waveguide configurations to assess the quality of the waveguide output beam, utilizing standard procedures [27] and the setup depicted in Figure 3b. In this arrangement, the plano-convex lens is moved horizontally, allowing for the measurement of the beam diameter in both the near and far fields. Consequently, using Equation (2), it is possible to determine the values obtained from M^2^, measured at a wavelength of 632 nm [28]. The experimental values were adjusted according to Equation (2), where d = 2*w* represents the diameter of the waveguide output beam measured at various focal distances (*z*). The beam diameter at the focus (*z*_0_) is given by d = 2*w*_0_.(2)$$d=d_{0}\sqrt{1+{\left(\frac{\left(M^{2}\lambda \left(z-z_{0}\right)\right)}{\left(\pi {d_{0}}^{2}\right)}\right)}^{2}}$$

The M^2^ at the output of a waveguide can decrease at longer wavelengths due to the increased likelihood of single-mode operation and the dominance of the fundamental mode. The international standard ISO 11146 [29] can be used to calculate M^2^ at 1064 nm, from Equation (3) using the values obtained from the setup in Figure 3b when *λ* ≪ *w*.(3)$$M^{2}=\frac{\pi \cdot \theta_{ideal}{{\cdot w}_{0}}_{ideal}}{\lambda }$$

In the equation above, $\theta_{ideal}$ represents the semi-angle of beam divergence measured in the far field, and ${w_{0}}_{ideal}$ denotes the waist radius at the beam focus. Additionally, polarization was measured using the experimental setup illustrated in Figure 4. The acquired data enabled the calculation of the percentage relationship between the power output values for each polarization axis. These results were crucial for evaluating polarization during the guidance process and its corresponding orientation.

To obtain top images of the waveguides and evaluate the structures formed by the fs laser, optical microscopy was utilized.

The visible optical absorption spectrum was obtained from 300 to 750 nm using a commercial spectrophotometer (OceanOpticsQE65 PRO, Orlando, FL, USA) to verify the influence of silver nanoparticles.

In order to investigate the nucleation of silver nanoparticles, a 200 kV transmission electron microscope (TEM) was used (Titan Cubed Themis, Thermo Fisher/FEI, Hillsboro, OR, USA). Scanning transmission electron microscopy (STEM) was performed to analyze the distribution of elements in the samples. To prepare the TEM sample, it was necessary to complete steps such as grinding, mixing with distilled water and partial decantation. Then, the isolated suspended particles were deposited on ultrafine copper grids coated with carbon film.

## 3. Results

With the aid of the experimental arrangement shown in Figure 3b it was possible to measure the beam waist (d = 2*w*) as a function of position z. Using Equation (2), it is possible to obtain the values of the M^2^ factor at 632 nm, for the horizontal and vertical axes, respectively, whose results are in Figure 5 for the straight waveguide configuration.

In Figure 6 and Figure 7, it is possible to observe the M^2^ values for the horizontal (Mx^2^) and the vertical axes (My^2^) for the S-bend architectures with 40 and 80 mm radius, respectively.

In the case of a Y-shaped architecture, it is possible to determine M^2^ for each arm, individually. The results of the first and second arms are presented in Figure 8 and Figure 9, respectively.

The results of M^2^ at 1064 nm were obtained by Equation (3) and are shown in Table 1 for all architectures as well as those at 632 nm. Using the setup of Figure 3a, it was possible to obtain the relative propagation losses, the output power ratio and the vertical polarization whose results are Table 1. The relative propagation loss was measured by comparing the S-bend and the Y configuration with the straight waveguide which has the lowest loss of all architectures; this can be attributed to the fact that the straight waveguide has the lowest path and leading to less losses. The output power ratio between the left and right arms of the Y waveguide was 50.47/49.53, suggesting potential applications in photonics.

Using optical microscopy and the setup of Figure 3a with the CCD camera instead of the power meter, it is possible to observe the modes of the waveguides; on the other hand, with the optical microscope, the structure of the waveguides can be seen. These results are shown in Figure 10 and Figure 11 for S-shaped waveguides with 40 and 80 mm radius, respectively. Figure 12 shows the top image of the Y configuration and the simultaneous mode of the two arms.

Results of the absorbance for samples with and without silver NPs can be seen in Figure 13; the absorbance shows intensity growth for wavelength smaller than 550 nm indicating the presence of a small concentration of silver NPs [12,30]. We attribute the absence of the absorption band associated with the localized surface plasmon modes to the small amount of silver NPs [31]. Although silver nanoparticles (Ag NPs) are synthesized during the annealing process, their concentration enhancement during fs laser irradiation cannot be excluded, as previously documented for tungsten lead–pyrophosphate glass [12,30]. During the fs laser processing, the nucleation of the NPs occurs due to the continuous heating of the focal region that promotes atomic mobility, leading to the enhancement of NPs growth [31].

Figure 14a,b present the images used to perform the nanoparticle size distribution. Figure 14c shows the TEM image of the sample doped with 2 wt% AgNO_3_ where it is possible to observe an isolated silver NP and the interplanar distance of 0.236 nm corresponding to the crystalline plane (1,1,1) of the face centered cubic silver [32,33,34]. Finally, in Figure 14d it is possible to see the size distribution of the silver NPs that were obtained from Figure 14a,b. The STEM images can be seen in Figure 15 and demonstrate uniform distribution of the chemical compounds on the glassy sample (Ge, Pb, O, Ag).

## 4. Conclusions

This work aimed to produce different structures of double waveguides using fs laser irradiation, with straight, S-bend and Y configurations, aiming at photonic applications, in GeO_2_-PbO glasses with silver nanoparticles. The use of the GeO_2_-PbO matrix as well as the configuration of double waveguides, each of them formed by 4 overlaps, was motivated by recent work of the group that demonstrated applications for optical amplifiers operating at 1064 nm, using Nd^3+^ ions as dopant, with and without metallic NPs.

Another motivation that deserves to be highlighted concerns the lack of literature regarding the configurations of the present work on glasses. In this work, the characterizations were made through propagation loss measurements, identification of the polarization and determination of the beam quality factor (M^2^). The glass samples were produced by the fusion method, followed by sudden cooling and appropriate heat treatment. For the S-bend waveguides, the curvature radii of 40 and 80 mm were used, with distance of 10 µm between the double waveguides.

The results with S-bend waveguides showed more suitable guidance for 80 mm radius when compared to the one with 40 mm as it presents greater symmetry between the guided axes. Thus, the best results for the quality factor (M^2^), at 632 and 1064 nm, and for the relative propagation losses were obtained for the 80 mm waveguide (Mx^2^ = 5.4, My^2^ = 5.2, at 632 nm, Mx^2^ = 3.2, My^2^ = 3.1 at 1064 nm and loss of 0.84 dB/cm). As reported in the literature, in less pronounced curvature radii, the losses are smaller, according to what was shown in the present work.

Using the same double waveguide configuration, a Y-waveguide was produced using an opening angle of 5° and a distance between the arms of 300 nm. A power ratio of 50.5/49.5 was verified, as well as the lowest relative loss propagation of 0.27 dB/cm, with respect to S-shaped waveguides. The similarity between the values of Mx^2^ = 6.1 and My^2^ = 4.1 for the first arm and Mx^2^ = 5.6 and My^2^ = 4.3 for the second one at 632 nm indicates that there is almost no difference in light guidance between them. This behavior was also observed at 1064 nm. It is important to highlight that vertical polarization was observed for all configurations of the present investigation.

S-bend waveguides have many applications and can be used to connect straight waveguides, alleviating the problem of horizontal displacement. They can also be used to build integrated waveguides, such as wavelength multiplexers, directional pairs, analog-to-digital converters, high-gain amplifiers and optical switches. The Y-shaped waveguides have applications in optical power dividers or optical power combiners in modulators and switches.

The present results show, for the first time, the possibility of producing passive waveguides with configurations formed by curves of different radii and in Y, from the glass matrix GeO_2_-PbO with silver nanoparticles, using the configuration of double waveguides formed, each of them, by 4 overlaps. Few glass compositions with silver nanoparticles have been studied so far (mainly phosphates) using the fs laser irradiation technique for the production of waveguides. The results show that the material is promising for applications in photonics that make use of the architectures addressed in this research and can be extended to oxide glasses of different compositions.

## Figures and Tables

**Figure 1 micromachines-16-00171-f001:**
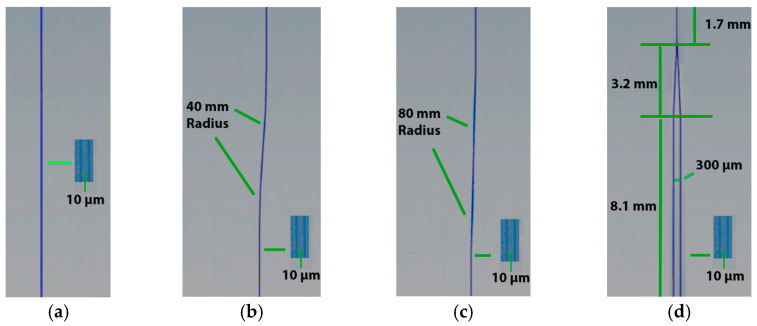
Illustration of waveguide architectures in GeO_2_-PbO glass irradiated with a femtosecond laser: (**a**) straight, S-bend with (**b**) 40 mm, (**c**) 80 mm radius and (**d**) Y-shaped.

**Figure 2 micromachines-16-00171-f002:**
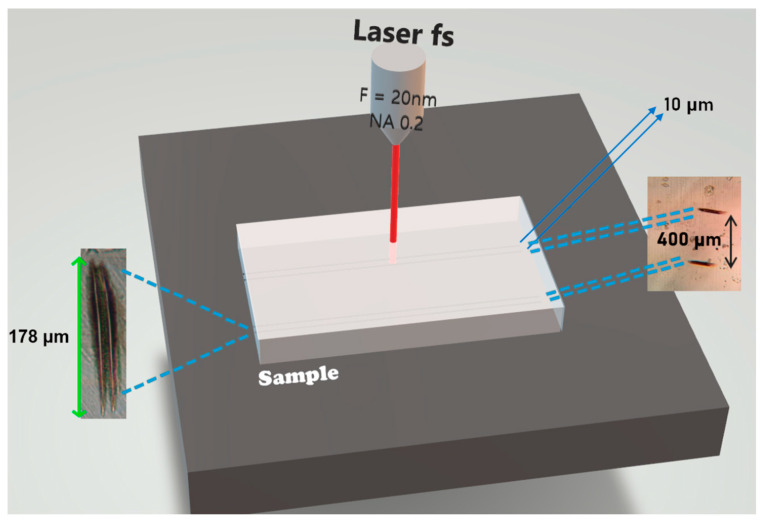
Illustration of the arrangement used for writing waveguides. Microscopy images of the damage caused by the laser at the time of writing on the entry and exit faces of the glass are shown.

**Figure 3 micromachines-16-00171-f003:**
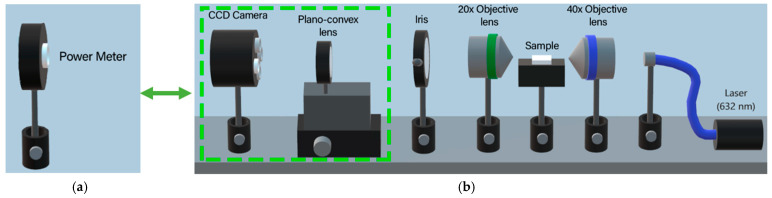
Experimental setup used to measure (**a**) propagation loss; (**b**) M^2^.

**Figure 4 micromachines-16-00171-f004:**
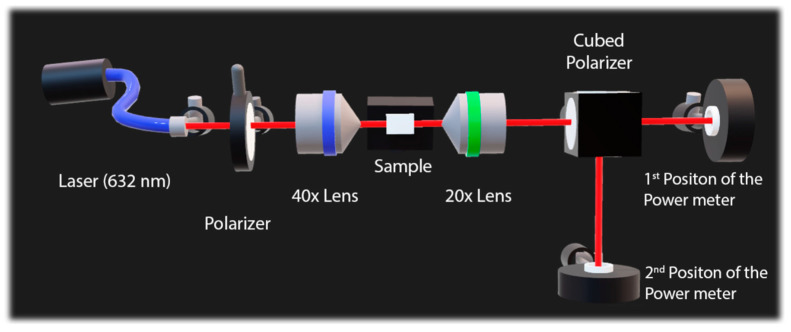
Experimental setup used for measurements of polarization of dual waveguides.

**Figure 5 micromachines-16-00171-f005:**
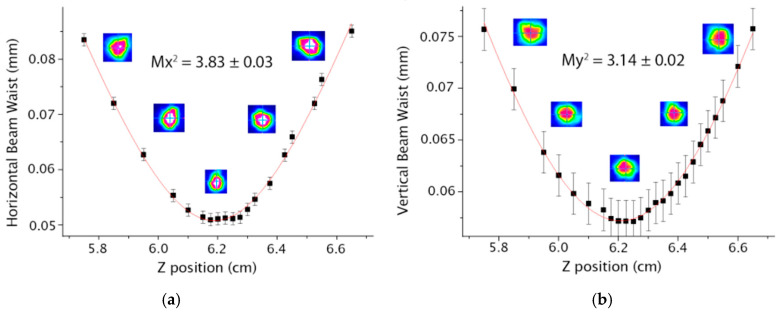
Results for the GP sample with a straight waveguide and a 10 µm separation between the guide walls (**a**) Mx^2^ (**b**) My^2^.

**Figure 6 micromachines-16-00171-f006:**
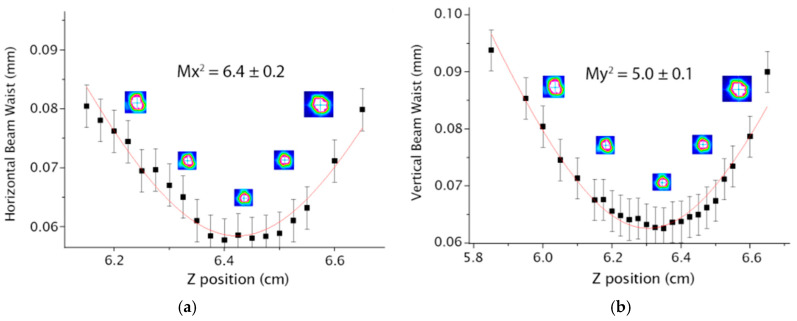
Results for the S-bend waveguide with 40 mm radius (**a**) Mx^2^ (**b**) My^2^.

**Figure 7 micromachines-16-00171-f007:**
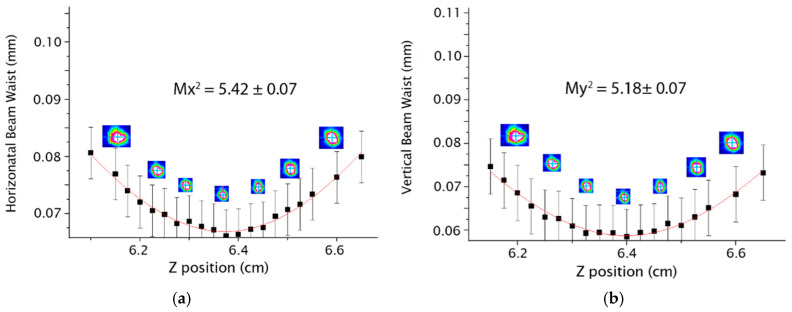
Results for the S-bend waveguide with 80 mm radius (**a**) Mx^2^ (**b**) My^2^.

**Figure 8 micromachines-16-00171-f008:**
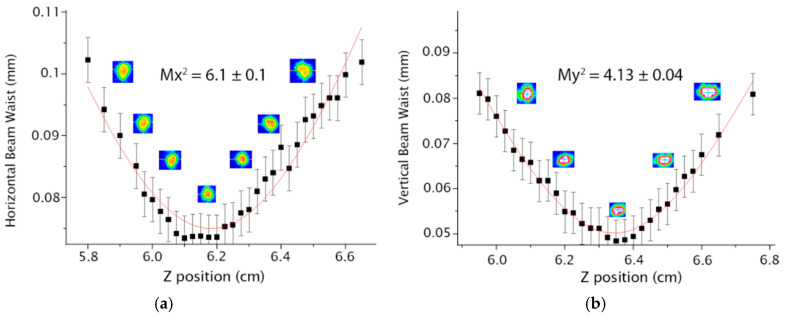
Results for the first arm of the Y-shaped waveguide (**a**) Mx^2^ (**b**) My^2^.

**Figure 9 micromachines-16-00171-f009:**
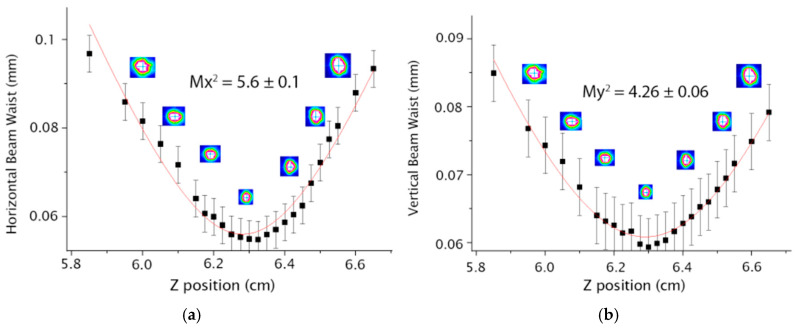
Results for the second arm of the Y-shaped waveguide (**a**) Mx^2^ (**b**) My^2^.

**Figure 10 micromachines-16-00171-f010:**
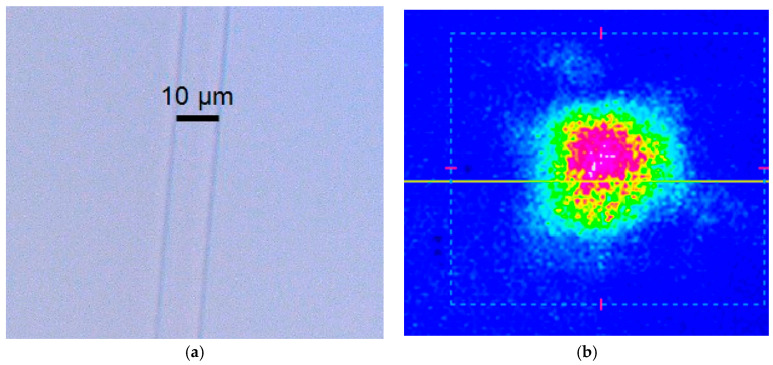
(**a**) Top view of the region of curvature of the S-bend waveguide with 40 mm radius. (**b**) View of the mode of the S-bend waveguide with 40 mm radius.

**Figure 11 micromachines-16-00171-f011:**
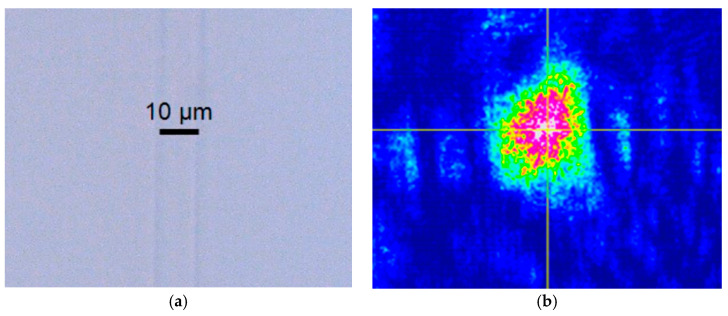
(**a**) Top view of the region of curvature of the S-bend waveguide with 80 mm radius. (**b**) View of the mode of the S-bend waveguide with 80 mm radius.

**Figure 12 micromachines-16-00171-f012:**
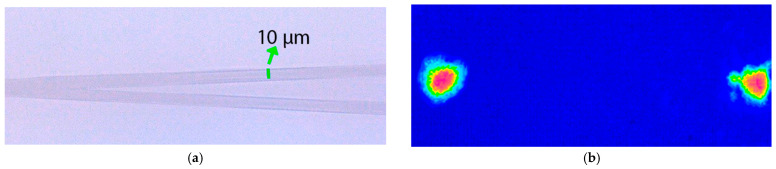
(**a**) Top view of the region of the aperture of the Y-shaped architecture. (**b**) Simultaneous view of the Y modes.

**Figure 13 micromachines-16-00171-f013:**
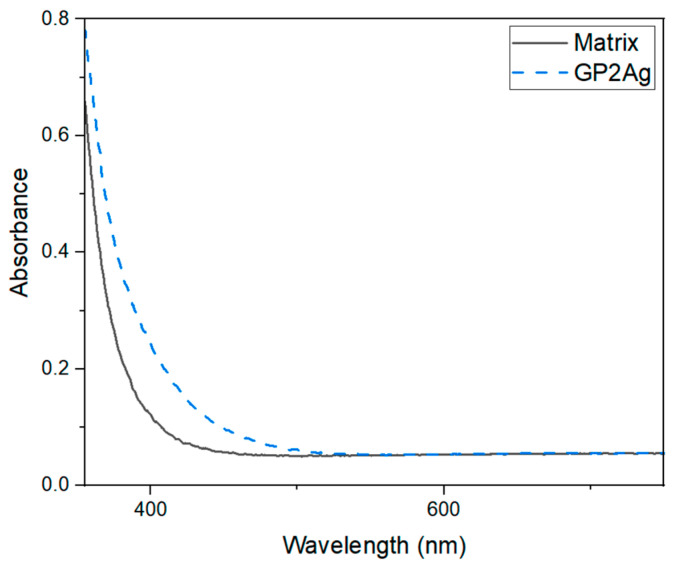
Absorbance in the visible range for GP samples with and without silver nanoparticles.

**Figure 14 micromachines-16-00171-f014:**
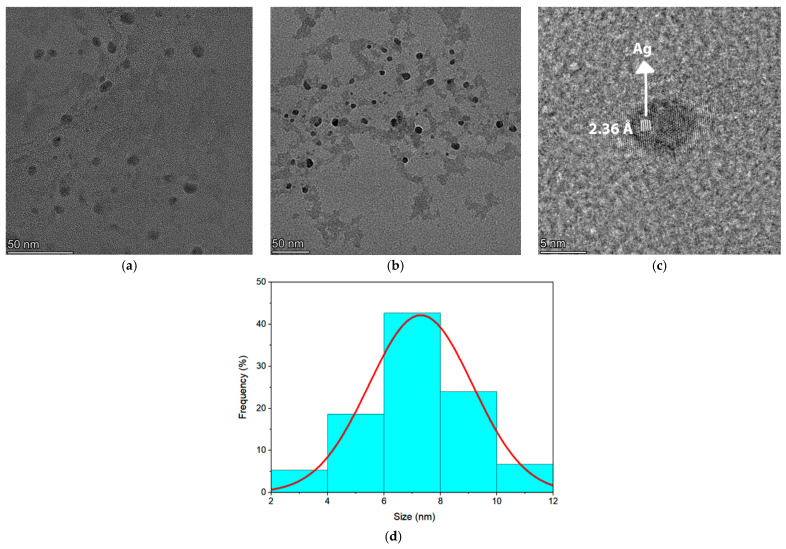
(**a**,**b**) TEM images of the GP2Ag sample with silver nanoparticles. (**c**) TEM image of an isolated silver nanoparticle with the interplanar distance corresponding to the crystalline plane (1,1,1) of a face centered cubic silver. (**d**) Size distribution of silver nanoparticles.

**Figure 15 micromachines-16-00171-f015:**
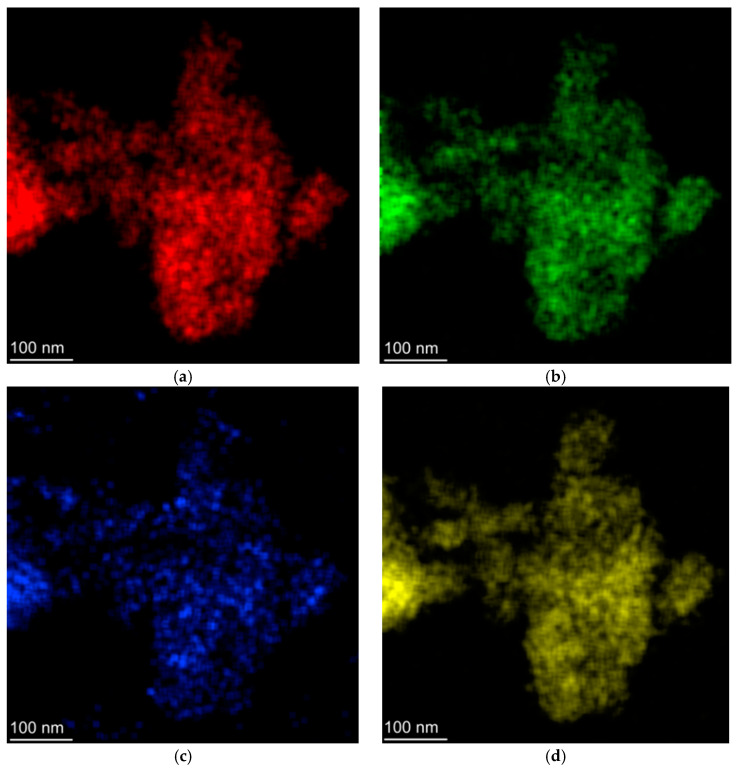
STEM images show the spatial distribution of (**a**) Ge, (**b**) Pb, (**c**) Ag and (**d**) O in the sample.

**Table 1 micromachines-16-00171-t001:** Results of quality factor, propagation losses and vertical polarization for all architectures.

Waveguide	Mx^2^ (632 nm)	My^2^ (632 nm)	Mx^2^ (1064 nm)	My^2^ (1064 nm)	Propagation Losses (dB/cm)	Vertical Polarization (%)
Straight	3.8	3.1	2.3	1.9	-	-
S-bend 40 mm	6.4	5.0	3.8	3.0	1.13	8.4
S-bend 80 mm	5.4	5.2	3.2	3.1	0.84	9.0
Y first arm	6.1	4.1	3.7	2.5	0.27	8.8
Y second arm	5.6	4.3	3.3	2.5		9.0

## Data Availability

The data presented in this study are available upon request from the corresponding author.

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
