# Peer review of "S-Bend and Y Waveguide Architectures in Germanate Glasses Irradiated by Femtosecond Laser"

_micromachines, 2025, doi:10.3390/mi16020171_

Round 1
Reviewer 1 Report
Comments and Suggestions for Authors
The manuscript titled “S bend and Y waveguide architectures in germanate glasses irradiated by femtosecond laser” by Fernandes et al. reports an experimental study on femtosecond laser writing of straight, S-bend and Y-shaped waveguides in germanium glasses and their characterization. Waveguides are written with a Ti:Sapphire laser and characterized using optical microscope, and TEM imaging. Near and far field profiles of the transmitted beam are obtained. The TEM images show the nucleation of nanoparticles after the laser exposure. The work has results of laser processing in new material systems and the results are interesting and proper to the journal. I recommend acceptance after the following issues are addressed.
1. Germanium shares some similarities as silicon, the latter of which has been used to write waveguides with different shapes (see, e.g., 10.2351/1.5139973). The authors should include and discuss those papers in their introduction to broaden the scope of the manuscript.
2. Please list the wavelength of the writing femtosecond laser beam.
3. Page 3: “... that show a decreased refractive index, ...” Magnitude of refractive index change needs to be listed and referenced.
4. Please explain the use of 45° polarization in waveguide writing.
5. Waveguide characterization is performed with 632 nm laser and the data are “converted” to 1064 nm using Eq. 3. This assumes other parameters (divergence angle, beam waist size) do not vary at different wavelengths, which may not be true. Please comment on the validity of Eq. 3 in converting data.
Author Response
Dear editor
We are returning our manuscript entitled: “S bend and Y waveguide architectures in germanate glasses irradiated by femtosecond laser” We would like to thank the reviewers for the comments and suggestions that really improved the paper. We attended all the suggestions of the reviewer and marked all the changes performed in red in the manuscript. We hope that we have clarified the issues raised by Reviewers.
Reviewer #1: The manuscript titled “S bend and Y waveguide architectures in germanate glasses irradiated by femtosecond laser” by Fernandes et al. reports an experimental study on femtosecond laser writing of straight, S-bend and Y-shaped waveguides in germanium glasses and their characterization. Waveguides are written with a Ti:Sapphire laser and characterized using optical microscope, and TEM imaging. Near and far field profiles of the transmitted beam are obtained. The TEM images show the nucleation of nanoparticles after the laser exposure. The work has results of laser processing in new material systems and the results are interesting and proper to the journal. I recommend acceptance after the following issues are addressed.
Comment 1: Germanium shares some similarities as silicon, the latter of which has been used to write waveguides with different shapes (see, e.g., 10.2351/1.5139973). The authors should include and discuss those papers in their introduction to broaden the scope of the manuscript.
Response 1: We agree with this comment and included the reference suggested as well as another one that are marked in the manuscript; we also included a comment about them in the last paragraph of the "Introduction"
Comment 2: Please list the wavelength of the writing femtosecond laser beam.
Response 2: We agree with this comment and informed the wavelength of the femtosecond laser beam (800nm) used for the writing in the second paragraph of Section 2.2 Waveguide Manufacture
Comment 3: Page 3: “... that show a decreased refractive index, ...” Magnitude of refractive index change needs to be listed and referenced.
Response 3: We agree with this comment and informed the magnitude of the refractive index change (10-3) in the first paragraph of Section 2.2 Waveguide Manufacture together with a reference
Comment 4: Please explain the use of 45° polarization in waveguide writing.
Response 4: We agree with this comment and explained the use of 45 polarization in waveguide writing in the second paragraph of Section 2.2 Waveguide Manufacture.
Comment 5: Waveguide characterization is performed with 632 nm laser and the data are “converted” to 1064 nm using Eq. 3. This assumes other parameters (divergence angle, beam waist size) do not vary at different wavelengths, which may not be true. Please comment on the validity of Eq. 3 in converting data.
Response 5: We agree with this comment and clarified this point by introducing a new reference for equation 3, as follows: international standard ISO 11146. We also included more explanation about M2 behavior at 1064 nm, in the forth paragraph of Section 2.3 Characterization.
Reviewer 2 Report
Comments and Suggestions for Authors
The manuscript explores the fabrication and characterization of S-bend and Y-shaped waveguide architectures in GeOâ‚‚–PbO glasses embedded with silver nanoparticles, using femtosecond laser irradiation. The study demonstrates how these waveguide configurations, along with the material properties of GeOâ‚‚–PbO glasses, are suited for photonic applications such as beam splitters and resonant rings.
The work provides useful insights into the performance of waveguides with varying geometries. For instance, S-bend waveguides with an 80 mm curvature radius exhibit improved beam quality and reduced propagation losses compared to tighter curvatures. The Y-shaped waveguides show balanced power distribution, indicating potential for applications in optical power dividers. The use of silver nanoparticles further enhances the optical properties, adding value to the study.
The manuscript effectively utilizes femtosecond laser technology for creating well-defined waveguide structures and provides detailed measurements of beam quality, propagation loss, and polarization. This study contributes to ongoing efforts in the field of integrated photonics, offering valuable data for optimizing waveguide architectures in heavy-metal oxide glasses. I recommend publication in Micromachines as is.
Author Response
Dear editor
Reviewer #2: The manuscript explores the fabrication and characterization of S-bend and Y-shaped waveguide architectures in GeOâ‚‚–PbO glasses embedded with silver nanoparticles, using femtosecond laser irradiation. The study demonstrates how these waveguide configurations, along with the material properties of GeOâ‚‚–PbO glasses, are suited for photonic applications such as beam splitters and resonant rings.
The work provides useful insights into the performance of waveguides with varying geometries. For instance, S-bend waveguides with an 80 mm curvature radius exhibit improved beam quality and reduced propagation losses compared to tighter curvatures. The Y-shaped waveguides show balanced power distribution, indicating potential for applications in optical power dividers. The use of silver nanoparticles further enhances the optical properties, adding value to the study.
The manuscript effectively utilizes femtosecond laser technology for creating well-defined waveguide structures and provides detailed measurements of beam quality, propagation loss, and polarization. This study contributes to ongoing efforts in the field of integrated photonics, offering valuable data for optimizing waveguide architectures in heavy-metal oxide glasses. I recommend publication in Micromachines as is.
Response 1: We thank the reviewer for the comments that motivate us to continue.